# Dietary diversity modification through school-based nutrition education among Bangladeshi adolescent girls: A cluster randomized controlled trial

Zannatun Nyma[1,2], Mahfuzur Rahman[1]*, Subhasish Das[1], Md Ashraful Alam[1], Enamul Haque[1], Tahmeed Ahmed[1]

1 icddr,b, Shaheed Tajuddin Ahmed Sarani, Mohakhali, Dhaka, Bangladesh, 2 Shiga University of Medical Science, Setatsukinowacho, Shiga, Japan

* mahfuzur.rahman@icddrb.org

## Abstract

### Objective

To measure the efficacy of school-based nutrition education on dietary diversity of the adolescent girls in Bangladesh.

### Methods

A matched, pair-cluster randomized controlled trial was conducted from July 2019 to September 2020. Randomization was done to select intervention and control schools. There were 300 participants (150 in the intervention and 150 in the control arm) at baseline. We randomly selected our study participants (adolescent girls) from grades six, seven, and eight of each school. Our intervention components included parents' meetings, eight nutrition education sessions, and the distribution of information, education, and communication materials. An hour-long nutrition education session was provided using audio-visual techniques in a class of intervention school once a week by trained staffs of icddr,b for two months. Data on dietary diversity, anthropometry, socio-economic and morbidity status, a complete menstrual history, and haemoglobin status of adolescent girls were collected at recruitment and after five months of intervention. We calculated the mean dietary diversity score of adolescent girls at baseline and at the endline. As the dietary diversity score was incomparable between the control and intervention arm at baseline, we performed the difference-in-difference analysis to assess the effect of the intervention.

### Results

Mean age of the adolescent girls was 12.31 years and 12.49 years in the control and intervention arms respectively. Percentages of consumption of organ meat, vitamin A-rich fruits and vegetables, legumes, nuts, and seeds were higher in the intervention arm than in the control arm at the end-line. The mean dietary diversity score remained unchanged in the control arm at 5.55 (95% CI: 5.34–5.76) at baseline and 5.32 (95% CI: 5.11–5.54) at the

**Data Availability Statement:** On the basis of recommendation of Institutional Review Board

(IRB), the Research Administration (RA) of icddr,b has imposed a restriction on disclosing any personal information of the patients or participants. However, data generated from icddr,b's research can be provided to interested researchers (recipients) for secondary data analyses upon approval of a Data Licensing Application & Agreement by the icddr,b Data Centre Committee. Interested personnel is recommended to consult this with icddr,b IRB Coordinator Mr. M A Salam Khan (salamk@icddrb.org).

**Funding:** As part of capacity building of young researcher, Mr. Mahfuzur Rahman received funding from icddr,b. The funders had no role in study design, data collection and analysis, decision to publish, or preparation of the manuscript.

**Competing interests:** The authors have declared that no competing interests exist.

endline. After the intervention, mean dietary diversity increased from 4.89 (95% CI: 4.67–5.10) at baseline to this mean was 5.66 (95% CI: 5.43–5.88) at the endline. Result from the difference-in-difference analysis revealed that the mean dietary diversity was likely to increase by 1 unit due to intervention.

## Conclusion

The shorter duration of the intervention in our study could not show whether it could change the behavior of adolescent girls in increasing dietary diversity through school-based nutrition education, but it showed a pathway for increasing dietary diversity at school. We recommend including more clusters and other food environment elements in retesting to increase precision and acceptability.

## Trial registration

This study was registered with ClinicalTrials.gov, trial registration no: NCT04116593. https://clinicaltrials.gov/ct2/show/NCT04116593.

## Introduction

The world's adolescent population (age of 10–19 years) is about 1200 million [1] and more than three-quarters of them are living in developing countries [2,3]. Bangladesh has a population of nearly 36 million adolescent people—more than one-fifth of Bangladesh's total population is between the ages of 10 and 19 [4]. According to the World Health Organization, children aged between 10–19 years are considered as adolescents [5]. The nutritional situation of youth, especially teenage girls in Bangladesh is in grievous state. Stunting among adolescents is 36% and low body mass index (BMI) 50% in Bangladesh [6]. Previous studies revealed that nutritional disorders affect a substantial percentage of adolescent girls to varying degrees [7,8].

There are some specific reasons for which pubescence is an intervention stage in the life cycle which is one of a kind [9]. After the first year of life, adolescence (10–19 years) is the second most critical period for physical growth [10,11]. During this period, adolescents gain up to 50% of their final adult weight, 20% or more than that of their adult height, and 50% of their adult skeletal mass [12]; of this entire growth, most is achieved during the early adolescence [13]. Study found that the peak velocity of linear growth of adolescent girls takes place approximately in six to twelve months prior to menarche [14]. In Bangladesh, the mean age for menarche is 12.8 years [15]. Therefore, any kind of health intervention will be very worthwhile if it is given during early adolescence (10–14 years). Furthermore, results from a multi-country study showed that maternal height is a key determinant of childhood nutritional status [16]. Since maternal height cannot be increased, we must go down the life cycle and consider increasing the height of adolescent girls at the population level. It would be the last favorable circumstance to interfere and annihilate the inter-generational vicious cycle of malnutrition [10,11].

Data collected from multiple countries revealed that nutrient inadequacy and inadequate dietary diversity are the major causes of adolescent malnutrition [17–19]. A diet containing diverse food items provides a wide range of macro and micronutrients and enhances the nutritional quality of the diet [20]. In contrast, starchy staples based on monotonous diets are deficient in important nutrients and subsidize to the burden of malnutrition [17,21]. Dietary diversity among the adolescents and women of Bangladesh is very low. Two-thirds (66%) of

women (including adolescents) consumed inadequately diversified diet (highest inadequacy in Rangpur, 72%) [22]. In order to improve adolescent nutrition, some efforts such as school gardening have been made by the Government of Bangladesh (GoB) through the Department of Agricultural Extension, Bangladesh Agricultural Research Council and its nutrition training institutions, notably the Bangladesh Institute of Research and Training in Applied Nutrition (BIRTAN). There are also nutrition modules in the curriculum of secondary school but are not compulsory for all students. Although the 'Home Economics' module includes a brief discussion of balanced diet, it does not cover dietary diversity focusing on 16 food groups recommended by the Food and Agricultural Organization (FAO). Therefore, it is very likely that adolescent girls are not receiving information on dietary diversity from schools.

Except for an intervention based on behavior change communication, the other programs focusing betterment of the nutritional status of adolescent girls have been found to be expensive and not scalable or sustainable [23,24]. Nutrition education has been defined as "any combination of educational strategies, accompanied by environmental supports, designed to facilitate voluntary adoption of food choices and other food and nutrition-related behaviors conducive to health and well-being; nutrition education is delivered through multiple venues and involves activities at the individual, community, and policy levels [25]. School children and adolescents pass their crucial period in the lifetime that determines their current and future behaviors. They are more likely to adopt and maintain health-promoting behaviors throughout their life when they are given these skills at this period [26]. School is a viable avenue to reach them in their adolescent period and nutrition education could be instrumental in increasing dietary diversity of them. However, there is scarcity of data regarding dietary diversity of Bangladeshi adolescent girls who are at their early adolescence and no studies have assessed the role of nutritional education in improving dietary diversity of school going adolescent girls. Considering all these contexts, we hypothesized that school-based nutrition education will improve dietary diversity among Bangladeshi adolescent schoolgirls.

We conducted a randomized controlled trial with an objective to measure the efficacy of school-based nutrition education on dietary diversity of the adolescent girls in Bangladesh.

## Methods and materials

The protocol for this trial and CONSORT checklist are available as supporting information; see S1 Checklist and S1 File.

### Study design and study population

We conducted a matched, pair-cluster randomized controlled trial from July 2019 to September 2020 in secondary girls' schools in Rangpur district, Bangladesh. We screened all secondary urban and rural girls' schools of Rangpur district on the basis of some criteria like— infrastructure of the school (building or tin-shed), presence of digital lab (laptop, projector), supply and availability of electricity, and number of students. We found that digital lab was available in all the girls' schools. In the case of urban schools, most of them were operated in buildings whereas the rural schools were operated in tin-shed houses. There was electricity in the urban schools but there were few disruptions in the supply of electricity in rural schools. The number of students was higher in the urban schools than that of the rural schools. In our study, we excluded the schools having any programme such as nutrition education, health message, mid-day meal. We maintained sufficient buffer zone between control and intervention schools. In rural site, the intervention and control schools were located in two different sub-districts. In the case of urban schools, even though intervention and control schools were located in Rangpur Sadar (same sub-district), they were located far enough from one another.

Our study participants were all the adolescent girls of 11–15 years from grade six, seven and eight of the selected schools. During assessment of the study participants, we followed inclusion criteria that included never married adolescent schoolgirls, girls having no chronic diseases, girls having no major psychiatric illness. Participant follow-up was completed in September 2020. After taking well-informed written consent from the mothers or caregivers of the study participants and assent from study participants, we approached for collecting data from them.

## Randomization

We visited 72 secondary girls' schools for screening. Based on inclusion criteria, we selected 28 schools out of 72. We prepared two separate lists—one for urban (6 urban schools) and one for rural schools (22 rural schools). From each list, clusters (schools) were paired based on similar criteria of the schools such as student number, the infrastructure of the schools, and availability of electricity. We randomly selected one pair from each list and within each pair, one school was assigned to the intervention arm and the other one was assigned to the control arm through randomization. Randomization was done by computer-generated random numbers using STATA. A statistician performed the randomization and the study investigators enrolled the participants. Although all the adolescent girls studying in grades six, seven and eight of the selected schools were our study participants, we collected data from randomly selected 25 students from each grade. There was a total of 75 students from each school. Thus, at baseline, there were a total of 300 study participants (150 students in the intervention and 150 students in the control arm). Due to the loss of follow-up, there were 293 participants in the endline survey (148 and 145 students in the intervention and control arm respectively). Marriage, migration, and reluctance to give permission to collect data due to the COVID-19 pandemic were three reasons for the loss of follow-up. "Fig 1" represents the flow diagram of the trial.

## Intervention components

We provided interventions including meeting with the parents of the adolescent girls, nutrition education sessions at schools and distribution of information, education and communication materials. We provided iron-folic acid supplementation according to the recommended dose of the Government of Bangladesh (GoB) in both the intervention and control arms. The recommended dose what we provided was once a week for consecutive 3 months, then 3 months of intervals, then again for 3 months. We also provided messages on water, sanitation, and hygiene in both arms. Detailed interventions under different components are described below:

**Parents' meeting.** At the beginning of the intervention, research team members arranged parents' meetings at the schools of the intervention arm. In the meeting, they informed the parents about the objective of the study, the benefits of dietary diversity (DD) on adolescent health, and how the parents can play role in ensuring the intake of diversified food for adolescent girls. In each meeting on average 150 parents attended.

**Nutrition education session.** We provided a total of eight nutrition education sessions at intervention schools for two months. An hour-long nutrition education session (60 minutes to 80 minutes long) was provided among the girls of each grade once in a week by the trained female staffs who received training on how to impart nutrition education among adolescent girls, from the investigators of the study. They consulted with headmasters and teachers at the schools to set the convenient time for providing education sessions. Nutrition education sessions were delivered using audio-visual techniques (audio-visual presentation). The components of eight education sessions have been described in detail in "Table 1".

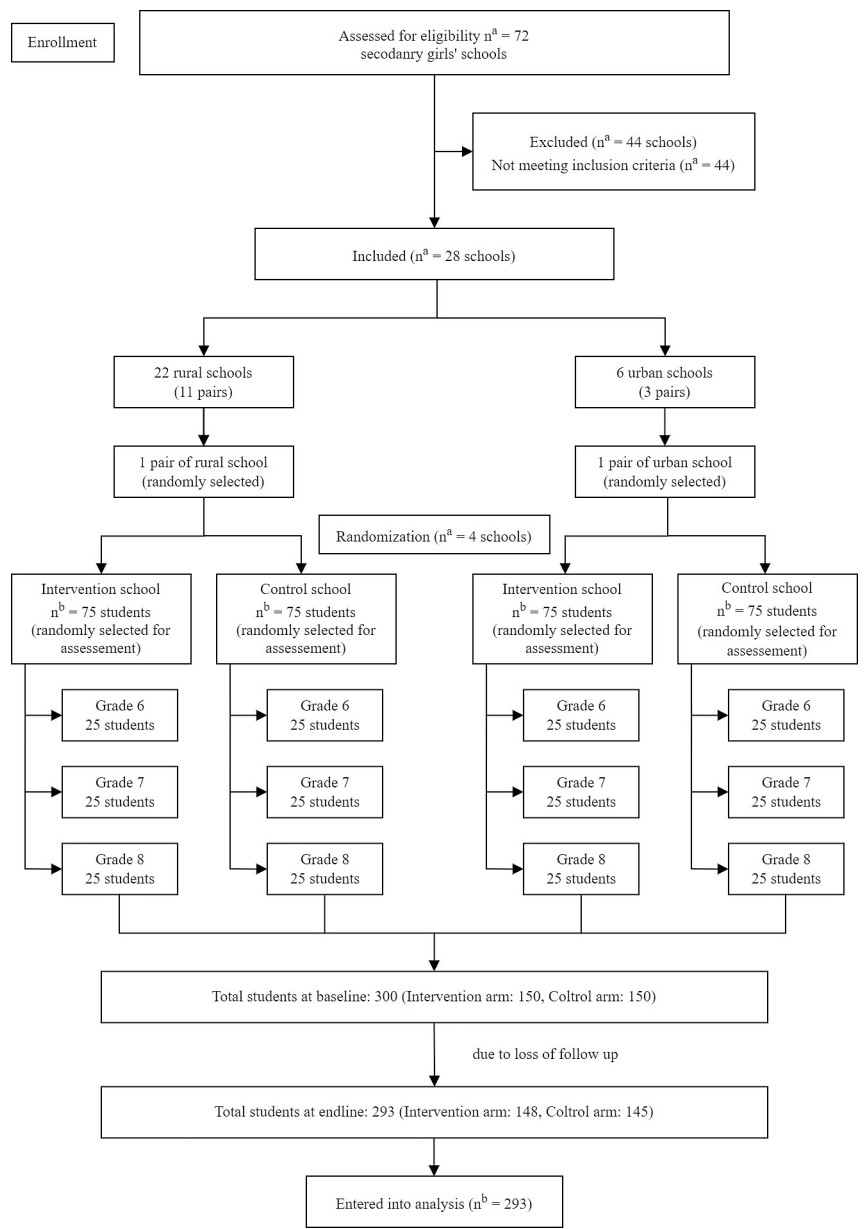

**Fig 1. Trial flow diagram.** (a) cluster (school). (b) participant.

These 8 nutrition education sessions were made based on Food and Agricultural Organization (FAO) recommended dietary diversity chart representing 16 food groups [27]. The First session included "Food group 1" which represented "Cereals", the second session was about "Food group 2" representing "White roots and tubers", the third session included "Food groups 3, 4, 5, 6, 7" which were about "Fruits and Vegetables", the fourth session was consisted of "Food group 8, 9, 10, 11" presenting "Animal protein", the fifth session was about "Food group 12" presenting "Plant protein", the sixth session included "Food group 13 and 14" which represented "Milk and milk products", seventh session was about "Food group 15" presenting "Sweets" and eighth session were about "Home-made food" incorporating the importance of consuming home-made food and health hazards of taking junk foods. In every

**Table 1. Components of our eight education sessions.**

| Sessions | Components |
|---|---|
| 1<sup>st</sup> session | • Basic food components<br>• Description on dietary diversity<br>• Overview on 16 food groups of "Guidelines for measuring household and individual dietary diversity"<br>• Detail description on "Food group 1 –Cereals" with pictures<br>• Benefits of "Cereals" on health |
| 2<sup>nd</sup> session | • Detail description on "Food group 2 –White Roots and Tubers" with pictures<br>• Benefits of "White Roots and Tubers" on health<br>• Health problems of absence of this food group in daily meal |
| 3<sup>rd</sup> session | • Conversation on "Vegetables and Fruits"<br>• Detail description on "Food group 3 –Vitamin A rich Vegetables and Tubers", "Food group 4 –Dark Green Leafy Vegetables", "Food group 5 –Other Vegetables" with pictures<br>• Detail description on "Food group 6 –Vitamin A Rich Fruits", "Food group 7 –Other Fruits" with pictures<br>• Benefits of "Vegetables & Fruits" (Food groups– 3,4,5,6,7) on health<br>• Health problems of absence of these food groups in daily meal |
| 4<sup>th</sup> session | • Conversation on "Animal Protein"<br>• Detail description on "Food group 8 –Organ Meat", "Food group 9 –Flesh Meats", "Food group 10 –Eggs", "Food group 11 –Fish and Sea Foods" with pictures<br>• Benefits of "Animal Protein" (Food groups– 8,9,10,11) on health<br>• Health problems of absence of "Animal Protein" in daily meal |
| 5<sup>th</sup> session | • Conversation on "Plant Protein"<br>• Detail description on "Food group 12 –Legumes, Nuts and Seeds" with pictures<br>• Benefits of "Plant Protein" (Food groups– 12) on health<br>• Health problems of absence of this food group in daily meal |
| 6<sup>th</sup> session | • Detail description on "Food group 13 –Milk and Milk Products" and "Food group 14 –Oils and Fats" with pictures<br>• Benefits of "Milk, Oils and Fats" on health<br>• Health problems of absence of these food groups in daily meal |
| 7<sup>th</sup> session | • Detail description on "Food group 15 –Sweets" and "Food group 16 –Spices, Condiments and Beverages" with pictures<br>• Importance of using "Spices and Condiments" in daily cooking |
| 8<sup>th</sup> session | • Importance of taking "Home-made foods"<br>• Health hazards of taking junk foods and beverages<br>• Strategies to select a diversified meal and remove monotony in food<br>• Proper timing and frequency of taking meals for 24 hours<br>• Selecting a diversified meal at low expense |

session, a detailed description of each food group was given by showing pictures of the locally available Bangladeshi food items of that food group to the students through power-point presentation. The health benefits of taking those 16 diversified food groups and health problems in absence of those food groups in daily meals were also explained. We conducted recap sessions and quiz in every week before starting the next session for evaluating whether the girls were learning effectively. Our intervention was to inform them about 16 diversified food groups in detail along with their health benefits, health problems in absence of those food groups in daily meals, proper timing as well as the frequency of taking meals in 24 hours, and how to select diversified meal at low cost.

**Information, education, and communication material (IEC materials.** The trained staff also distributed posters and pamphlets containing the messages of dietary diversity among the students at schools belonging to the intervention arm. Posters and pamphlets containing Food and Agricultural Organization (FAO) recommended dietary diversity chart represented 16 food groups [27] (Picture of our poster containing FAO recommended 16 food groups representing dietary diversity is attached as S1 Appendix). Local Bangladeshi foods were also included in the posters and pamphlets. Our staff suggested the students to hang the posters in front of their dining tables where all members of the households used to sit to take daily meals

or in a kitchen where a mother or caregiver used to cook foods for household members. They also suggested the students to keep pamphlets in their school bags and consult among themselves about the dietary diversity chart which would help them remember the 16 dietary diversity food groups.

## Trial outcomes

The primary outcome of our trial was individual-level dietary diversity among adolescent girls. Dietary Diversity is defined as the number of different foods or food groups consumed over a given reference period [28]. The secondary outcomes of interest were household-level dietary diversity of the adolescent girls, changes in anthropometry, and anaemia status. However, in this manuscript, we have reported the results on the primary outcome and the results from the qualitative component are not presented here.

## Sample size

We calculated sample size based on the effect size of 20 percentage point using the formula of sample size estimation for conventional cluster randomized control trial. After considering the design effect of 1.5, 80% power, and 5% attrition rate, the desired sample size was 148 in each arm. Since we had two schools in each arm, our estimated sample from each school was 74 girls. Thus, to reach the desired sample size we collected data from 25 girls from each grade in a school (a total of 75 from each school and 150 from each arm).

## Data collection

Data on dietary diversity, anthropometry, socio-economic and morbidity status, menstrual history, and haemoglobin status of adolescent girls were collected at baseline and after five months of intervention. Dietary diversity data were collected using 24-hour recall dietary diversity questionnaire. Face-to-face interviews were conducted to collect data from adolescent girls at schools. We also collected data on socio-economic status, household dietary diversity, water, sanitation, and hygiene from the caregivers of adolescent girls at the household level. Data collection at the school and household level was done during baseline and at the endline survey.

## Statistical analysis

Data entry was done by using Microsoft Access and data analysis was done by STATA (version 14). Dietary diversity score was determined by adding the number of food categories consumed by each participant over the previous 24 hours. The questionnaire contained 16 food groups. The 16 food groups were—cereals; white roots and tubers; vitamin A-rich vegetables and tubers; dark green leafy vegetables; other vegetables; vitamin A-rich fruits, other fruits; organ meat; flesh meats; eggs; fish and seafood; legumes; nuts and seeds; milk and milk products; oils and fats; sweets, spices, condiments, and beverages. We combined certain food groups into a single food group for analytical purposes. We analyzed 9 food groups for measuring individual dietary diversity score. Dietary diversity score ranged from 0 to 9. According to the guideline, when we aggregated two food groups into one, any of the food group's "Yes" answer is considered as "Yes" for that aggregated group and numbered as "1" for the respected group. According to the guidelines, there is no established cut-off point in terms of food groups to indicate adequate or inadequate dietary diversity [27]. For this, we calculated the mean dietary diversity score for analytical purpose.

Categorical variables were presented as frequency and percentage. Continuous variables were presented as mean with standard deviation. To see the relationship with the study group, we performed t-test for normally distributed data and Mann-Whitney test for skewed data. As the dietary diversity score was incomparable between the control and intervention arm at baseline, difference-in-difference (DID) analysis was performed to assess the effect of the intervention. During performing the DID we adjusted for adolescent's age, adolescent girls' father's age, years of schooling of caregiver, adolescents' father's years of schooling, household's monthly income, household head's occupation and asset index.

### Ethical considerations

Institutional review board of International Centre for Diarrhoeal Disease Research, Bangladesh (icddr,b) consisted of Research Review Committee and Ethical Review Committee reviewed and approved the study protocol. Screening and enrolment of the study participants were done at the study site (selected urban and rural schools of Rangpur). Since our intervention was school-based nutrition education, we obtained initial approval from the school authority and headmaster. If the legal guardians (school headmasters) and adolescent girls were interested to volunteer in the study, the designated staffs proceeded to screening and consenting. As our study participants were <18 years, we took **written** assent from the adolescent girls and **written** consent from the mother or caregiver of the adolescent girls by visiting their households before including them in this study.

### Results

A total of 300 adolescent girls from four schools (2 urban and 2 rural schools) participated in this study. "Table 2" demonstrated that the mean age of the participants was 12.31 years and 12.49 years in the control and intervention arms respectively. The mean years of schooling of the caregivers of the participants were 6.07 years in the control and 5.11 years in the intervention arm. The median household monthly income was 180 USD in the control arm and 120 USD in the intervention arm. Most of the household heads' occupations were businessman (45%) followed by agriculture (27%), government and private service (14%) in the control arm, but in the intervention arm most of the household heads' occupations were agriculture (38%) followed by businessman (16%) and government and private service (11%). In the control arm, most (29%) of the households of the adolescent girls belonged to the richest wealth quintile whereas in the intervention arm, most (31%) of the households were from the poorest quintile.

"Fig 2" illustrated that all the adolescent girls interviewed consumed starchy staples, both in baseline and in end-line regardless of intervention and control arm. In both arms, 99% of adolescent girls consumed other fruits and vegetables at baseline and end-line and this percentage reached to 100% for both groups at the endline.

"Fig 2" represented that a significant change was noticed in the consumption of organ meat. In the control arm, the consumption of organ meat remained almost unchanged at 13% at baseline and 12% in end-line respectively but in the intervention arm, it was 13% at baseline and 32% in endline. We found that consumption of other vitamin A-rich fruits and vegetables were 25% at baseline and 16% in the end-line in the control arm. In the intervention arm, it increased from 16% in baseline to 33% in end-line. We revealed that consumption of legumes, nuts and seeds among the adolescent girls in the intervention arm increased from 46% at baseline to 70% at endline. Consumption of milk and milk products remained unchanged in the control arm across the surveyes-53% in baseline and 57% in endline; it increased from33% in baseline to 61% in end-line in the intervention arm. However, consumption of dark green

**Table 2. Basic characteristics of the study participants.**

| Variables | Control | Intervention | p-value |
|---|---|---|---|
| Age of the adolescent girls (Mean ± SD[a]) | 12.31 ± 1.18 | 12.49 ± 1.38 | 0.219 |
| Age of the caregiver (Mean ± SD) | 38.4 ± 7.47 | 38.13 ± 8.51 | 0.766 |
| Age of the father (Mean ± SD) | 43.87 ± 6.44 | 42.19 ± 7.39 | 0.039 |
| Years of schooling of the caregiver (Mean ± SD) | 6.07 ± 4.09 | 5.11 ± 3.98 | 0.040 |
| Years of schooling father (Mean ± SD) | 6.27 ± 4.72 | 5.12 ± 4.83 | 0.040 |
| Household monthly income in USD, Median (IQR[b]) | 180 (120, 240) | 120 (96, 204) | 0.003 |
| Household monthly expenditure in USD, Median (IQR) | 144 (96, 216) | 120 (84, 180) | 0.009 |
| Household size (Mean ± SD) | 5.1 ± 1.86 | 4.82 ± 1.43 | 0.145 |
| **Occupation of the caregiver, n (%)** | | | |
| Housewife | 141 (94) | 138 (92) | 0.422 |
| Service | 6 (4.00) | 5 (3.33) | |
| Others (Unemployed, Business, Farming, Student) | 3 (2.00) | 7 (4.67) | |
| **Occupation of the household head, n (%)** | | | |
| Business | 68 (45.33) | 25 (16.67) | 0.000 |
| Agriculture | 41 (27.33) | 57 (38.00) | |
| Service | 21 (14.00) | 17 (11.33) | |
| Others (Day labour, Auto Driver etc.) | 19 (12.67) | 44 (29.33) | |
| Unemployed | 1 (0.67) | 7 (4.67) | |
| **Asset Index, n (%)** | | | |
| Poorest | 18 (12) | 47 (31.33) | |
| Poor | 24 (16) | 31 (20.67) | |
| Middle | 31 (20.67) | 33 (22) | |
| Rich | 34 (22.67) | 23 (15.33) | |
| Richest | 43 (28.67) | 16 (10.67) | |
| **Religion, n (%) Fisher Exact** | | | |
| Islam | 146 (97.33) | 140 (93.33) | 0.085 |
| Hindu | 3 (2) | 10 (6.67) | |
| Buddhist | 1 (0.67) | 0 (0) | |

[a] Standard deviation

[b] Inter quartile range.

leafy vegetables significantly decreased from 68% in baseline to 37% in endline and 49% in baseline to 37% in endline, in the control and intervention arms respectively.

"Fig 3" presented that the mean dietary diversity score was 5.55 (95% CI: 5.34–5.76) in the control arm and 4.89 (95% CI: 4.67–5.10) in the intervention arm (p<0.001) at baseline but at end-line, this mean score was 5.32 (95% CI: 5.11–5.54) in the control arm and 5.66 (95% CI: 5.43–5.88) in the intervention arm (p<0.033).

Due to differences in the mean dietary diversity score between the control and intervention group at baseline, we performed difference in difference (DID) analysis and "Table 3" revealed that the mean dietary diversity score among the adolescent girls was likely to be increased by 1 unit due to intervention. It indicates that the intervention has potential to improve nutrient adequacy of adolescent girls as the dietary diversity score is considered to be a proxy indicator of nutrient adequacy.

## Discussion

Our study demonstrated that school-based nutrition education has the potential to increase dietary diversity among adolescent girls. It indicates that along with the regular curriculum in

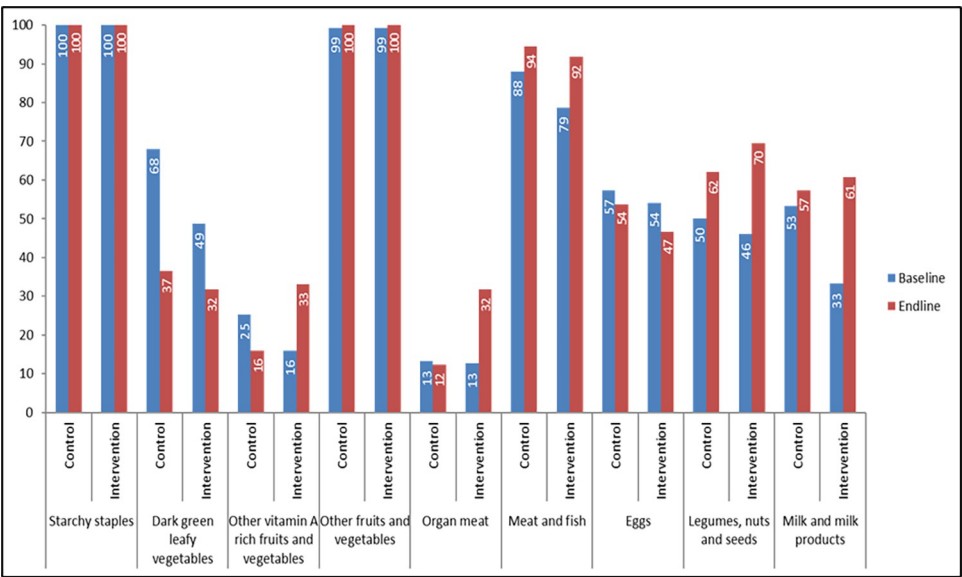

**Fig 2. Dietary diversity (food groupwise) among the adolescent girls.**

secondary schools, nutrition education can be added on to improve the dietary diversity of adolescent girls which corresponds with the findings of other studies [26,29,30]. However, the modalities of the interventions that targeted adolescent girls to improve their dietary diversity might differ in terms of the activities, mode of delivery of interventions, and contexts.

Nutrition education is a much more comprehensive enterprise than nutrition information dissemination [25]. Nutrition education is a method by which nutrition information is delivered to the target population. Successful school-based health and nutrition education interventions encompass theory-driven strategies; skills-building with adequate time and intensity to

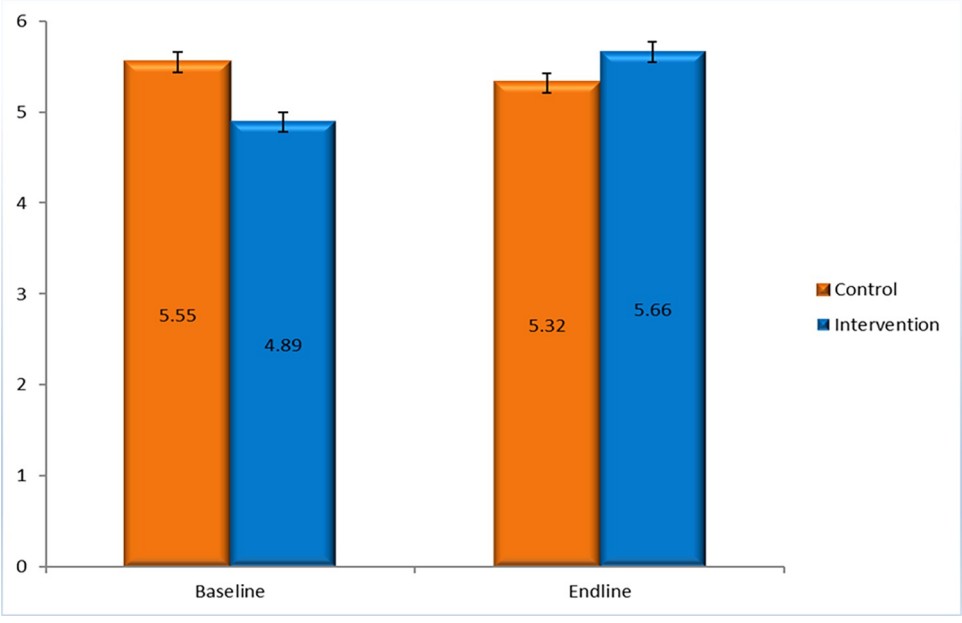

**Fig 3. Dietary diversity score (mean) among the adolescent girls.**

**Table 3. Changes in dietary diversity score due to nutrition education.**

|  | Unadjusted coefficient (95%CI) | P-value | Adjusted coefficient (95%CI) | P-value |
|---|---|---|---|---|
| Intervention effect | 0.985 (0.615, 1.356) | <0.001 | 1.003 (0.626, 1.380) | <0.001 |

influence attitudes and achieve the development of skills; with adequate teaching methods and teacher training opportunities; and with cultural relevance and family involvement. When school health and nutrition programs are connected to the surrounding community, not only do students benefit but also school personnel, students' families, and the entire neighborhood [26,31]. Along with providing nutrition education session, we also conducted parent's meeting as part of our intervention and conducted periodic meetings with school teachers and headmasters to increase compliance of the intervention.

A study done in Ethiopia provided school-based health and nutrition education using a combination of strategies including peer-groups, school media, and health clubs, family and community participation. It showed there was a significant increase of the proportion of school children consuming diversified diet due to intervention. The positive change in dietary diversity in this study [26] can result from the comprehensiveness of the intervention strategies and duration of intervention. It is also evident that not only in low- and middle-income countries, but also in the context of upper-middle-income countries like Iran, school-based nutrition education along with other comprehensive interventions can have impact on the dietary diversity of adolescent girls. In Iran, the study included nutrition workshops, interactive seminars as intervention with some other components with the aim of increasing dietary diversity among adolescent girls [29]. They found that dietary behaviours of adolescent girls including intake of fruits and vegetables, low-fat dairy, fats and oils, and less consumption of junk foods had been noticed in the intervention group. Our study used only audio-visual aid to promote dietary diversity for a shorter duration at school settings and it did not have such a comprehensiveness as the above-mentioned studies did. Because in the context of low- and middle-income countries, considering the limited resource and above all the sustainability of the intervention and it's scaling up at the national level, we wanted to establish a sustainable model of school-based nutrition education for improving dietary diversity among adolescent girls. Scaling up of an innovative intervention and its sustainability is essential for an effective and longer-term impact of the intervention [32].

In low- and middle-income countries, the school feeding program (SFP) has been found to be instrumental to increasing dietary diversity of adolescent girls but such a program has some limitations including irregular supply and storage of materials (i.e., food items) and loss of educational time at school [23]. Withdrawn of direct food under the school feeding program (SFP) is very likely to have a negative impact on the sustainability of the program [33]. Our study has focused on behavior change of adolescent girls through nutrition education at school setting, therefore, once the intervention is effective it is very likely to be sustainable. However, shorter duration of the intervention in our study could not demonstrate whether it could change behavior of adolescent girls in increasing dietary diversity through school-based nutrition education, rather it showed a pathway on how to increase dietary diversity at school setting.

Due to our intervention, consumption of some food groups was increased among intervention arm than in control arm in endline. Findings of our study showed that 100% of adolescent girls in the baseline as well as in the end-line consumed starchy staples both in intervention and in the control group. It might be due to the fact that starchy staples are culturally and socially acceptable diets in our country. It is the main food source and most available diet in

Bangladesh just like other developing countries [34] but monotonous staple diets lack essential nutrients which lead to macro and micronutrient deficiencies among vulnerable groups of population like adolescents [35,36]. Similar findings have been seen in a cross-sectional survey conducted in Gondar city, northwest Ethiopia among adolescent girls. They found that the majority (97.7%) of adolescent girls consumed starchy staples [30]. Likewise, another study conducted in a rural community of Eastern Ethiopia among 10–19 years of adolescent girls revealed that the most commonly consumed food group by study participants was starchy staples (97.2%) while the least utilized food group was organ meat (12.9%) [37].

We also observed in our study that the consumption of organ meat increased from 12% at baseline to 32% at the endline in intervention but it remained unchanged in the control arm across the surveys. It can be assumed that awareness of organ meat and its benefit can foster the consumption of organ meat. The same reasons could be applicable to increasing the consumption of legumes, nuts and seeds, and milk and milk products. A study done in Southern Ethiopia found that the pulse and legumes intake of the beneficiary households was substantially higher (91.7%) compared with the non-beneficiaries (8.3%) which clearly indicated the effect of school-based nutrition intervention [23]. However, unexpectedly, we found that intake of dark green leafy vegetables was lower in both the intervention and control arm and it might result from seasonality. We collected our baseline data in the month of January (winter season), when there was an abundance of dark green leafy vegetables in Bangladesh but when we performed our end-line data collection, it was the month of September, which is known as the flood-prone season during which dark green leafy vegetables were not readily available. A previous study examining of the seasonal patterns of food intake, growth and prevalence of malnutrition revealed that leafy vegetables intake was found to be lowest in late October and early November which corresponds our finding, and both months are too close to our endline data collection time (September) [38]. Moreover, the covid-19 pandemic hit Bangladesh in March 2020 and from mid-March to August the country was under lockdown. So, there was an issue of food insecurity [39] and that could contribute to the food consumption of our study area as well.

A study done in Southeast Asia found that a considerable proportion of girls did not eat eggs, milk or dark green leafy vegetables but a larger proportion of them consumed meat and fish [40]. We found similar findings that a considerable proportion of adolescent girls consumed a higher proportion of meat and fish irrespective of intervention and control arm across the surveys.

EAT-Lancet commission emphasized a global planetary health diet that is healthy for both people and planet. Whole grains, fruits, vegetables, nuts and legumes comprise a greater proportion of foods consumed. Meat and dairy constitute important parts of the diet but in significantly smaller proportions than whole grains, fruits, vegetables, nuts and legumes [41]. Due to our nutrition education intervention, consumption of "other vitamin A-rich fruits and vegetables" increased almost double in endline than in baseline among intervention group which we can consider as an appreciable finding. Along with this, "legumes, nuts and seeds" food group consumption was also increased in intervention arm in endline which is another appreciable finding because nuts and seeds are nutrient-dense (contain unsaturated fatty acids, fibre, vitamins, minerals, antioxidants, and phytosterols) and nut consumption reduces blood lipid concentrations, oxidative stress, inflammation, visceral adiposity, hyperglycaemia, and insulin resistance [41]. Although excessive intake of animal protein has some negative health impact in later life like chance of developing cardio-vascular diseases (specially for red meat consumption), EAT-Lancet commission also suggested that protein quality (defined by effect on growth rate) reflects the amino acid composition of the food source, and animal sources of protein are of higher quality than most plant sources. High-quality protein is particularly important for

growth of infants and young children, and possibly in older people losing muscle mass in later life [41]. Due to our intervention, consumption of "organ meat" was almost double in endline compared baseline among intervention group.

The strength of our study is it is the first paired-matched randomized control trial in Bangladesh that aimed to measure the efficacy of school-based nutrition education in improving dietary diversity among adolescent girls. Nutrition education was provided in a classroom setting and the participation of the students was ensured by informing them prior to the day of the nutrition education session in a week, thus the fidelity of the interventions was ensured. This study used audio-visual aids for delivering the intervention. Since the audio-visual aids are available in the secondary school settings [42] in Bangladesh, the study shows the pathway to scale up within the resources available at schools.

There are some limitations of our study which worth to be mentioned. Firstly, due to budget constraints, we conducted our study in a limited number of clusters (schools) which is likely to affect the methodological rigor of the study. Secondly, due to the COVID-19 pandemic, we had to provide intervention for two months instead of three months. We also had to forgo the midline survey because the whole country was in lockdown state. Thirdly, the increase of 1 unit in the mean score of dietary diversity in the intervened group does not have clinical significance. However, mean dietary diversity score was increased only 1 unit in intervention arm compared to control arm, it was statistically significant even after all other variables were adjusted. Moreover, this 1 unit increase in the mean score of dietary diversity could be minimal, but significant within a shorter period and with many challenges, the results indicated increase of some food groups in the intervention group. Our baseline survey was conducted in January 2020 when there was no COVID-19 pandemic situation. As we used the same questionnaire in baseline and endline, so we could not collect any data representing the effect of the COVID-19 pandemic on food availability and access. The absence of data regarding seasonality was one of the limitations of our study. We could not capture the effect of seasonality on dietary diversity since the data were not collected at the different time points (seasons) over the year. We could not collect data regarding indicators of eating behavior disorders because we had to make our questionnaire brief and short to complete the interview with the adolescent girls within a limited time (some particular times from school hours). In addition, we did not incorporate any evidence-generating tool other than quiz (quiz and recap session were taken before starting the next session) by which we could measure their knowledge of the given intervention.

## Conclusion

The shorter duration of the intervention in our study could not demonstrate whether it could change the behavior of adolescent girls in increasing dietary diversity through school-based nutrition education, instead it showed a pathway on how to increase dietary diversity at school setting. With a limited resource, our school-based nutrition education intervention in real settings of the schools has demonstrated its feasibility in improving dietary diversity of the adolescent girls. Since this study was conducted in the COVID-19 situation with many other challenges and limitations it recommends to retest this intervention before scaling up in the larger scale. In retesting, we recommend including more clusters and other elements of the food environment to increase its precision and acceptability.

## Supporting information

**S1 Checklist. CONSORT checklist.**
(PDF)

**S1 File. Study protocol.**
(PDF)

**S1 Appendix. The poster contains 16 food groups representing dietary diversity.**
(PDF)

**S2 Appendix. The output of DID with adjusted variables.**
(TIFF)

## Acknowledgments

We gratefully acknowledge Dr. S. M. Tafsir Hasan and Md. Tariqujjaman who provided their valuable feedback during preparation of this manuscript and data analysis. We thank the study participants, field staffs and especially the school authorities and teachers at the selected schools for their endless support during our research.

Current donors providing unrestricted support include the Governments of Bangladesh, Canada, Sweden and the UK. We gratefully acknowledge our core donors for their support and commitment to icddr,b's research efforts.

## Author Contributions

**Conceptualization:** Zannatun Nyma, Mahfuzur Rahman, Tahmeed Ahmed.

**Data curation:** Md Ashraful Alam, Enamul Haque.

**Formal analysis:** Mahfuzur Rahman, Subhasish Das, Md Ashraful Alam, Enamul Haque.

**Funding acquisition:** Tahmeed Ahmed.

**Investigation:** Zannatun Nyma, Mahfuzur Rahman, Subhasish Das, Tahmeed Ahmed.

**Methodology:** Zannatun Nyma, Mahfuzur Rahman, Subhasish Das.

**Project administration:** Zannatun Nyma, Mahfuzur Rahman.

**Software:** Md Ashraful Alam, Enamul Haque.

**Supervision:** Mahfuzur Rahman, Subhasish Das.

**Visualization:** Md Ashraful Alam, Enamul Haque.

**Writing – original draft:** Zannatun Nyma.

**Writing – review & editing:** Mahfuzur Rahman, Subhasish Das, Tahmeed Ahmed.

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
