## [Decision Letter · Decision Letter 0]

8 Aug 2022

PONE-D-21-39192Dietary Diversity Modification through School-Based Nutrition Education among Bangladeshi Adolescent Girls: A Cluster Randomized Controlled TrialPLOS ONE

Dear Dr. Rahman,

Thank you for submitting your manuscript to PLOS ONE. After careful consideration, we feel that it has merit but does not fully meet PLOS ONE’s publication criteria as it currently stands. Therefore, we invite you to submit a revised version of the manuscript that addresses the points raised during the review process.

Please see the comments from two reviewers below. They have raised several queries on the presentation of the methodology and statistics, which we ask that you address.

We look forward to receiving your revised manuscript.

Kind regards,

Hanna Landenmark

Staff Editor

PLOS ONE

Journal Requirements:

Reviewers' comments:

Reviewer's Responses to Questions

**Comments to the Author**

1. Is the manuscript technically sound, and do the data support the conclusions?

Reviewer #1: Partly

Reviewer #2: No

2. Has the statistical analysis been performed appropriately and rigorously? 

Reviewer #1: No

Reviewer #2: No

3. Have the authors made all data underlying the findings in their manuscript fully available?

Reviewer #1: Yes

Reviewer #2: Yes

4. Is the manuscript presented in an intelligible fashion and written in standard English?

Reviewer #1: Yes

Reviewer #2: Yes

5. Review Comments to the Author

Reviewer #1: The sample size needs rationale based on intra class correlation in the cluster environment. They state that they calculated sample size based on the effect size of 20 percentage point using the formula of sample size estimation for conventional cluster randomized control trial. They do not state the anticipated power based on this approach.

The main result is seen in Table 3 using the DID which revealed that the mean dietary diversity score among the adolescent girls was likely to be increased by 1 unit due to intervention. The authors give no explanation as to the clinical significance of this difference of one unit. They do, in fact, discuss some limitations of the study seen in the discussion section.

They did state that during performing the DID they adjusted for adolescent’s age, adolescent girls’ father’s age, years of schooling of caregiver, adolescents’ father’s years of schooling, household’s monthly income, household head’s occupation and asset index. This data should be presented as to the significance of these factors, if any.

Reviewer #2: General comments:

The aim of this study was to measure the efficacy of school-based nutrition education on dietary diversity of the adolescent girls in Bangladesh. The paper is interesting, but the measuring or evaluating the effectiveness of nutrition education is something very compromising, especially because the type of nutrition education or the pedagogical basis are not specified. The execution period of the community trial is striking, which was shortly before the COVID-19 pandemic began and the study ended in the total closure of schools (September 2020), which had an effect on the results, and therefore affects the conclusion reached with this research.

Specific comments:

INTRODUCTION

- In the introduction, "nutritional education" is not defined and the theoretical references of the educational intervention are not established.

- The hypothesis of the investigation is not justified with valid antecedents.

MATERIALS AND METHODS:

Study design and study population

Within its inclusion criteria, indicators of eating behavior disorders are not mentioned, which is more frequent in adolescent women.

Specify the characteristics of urban and rural schools, and the contexts that could have defined a lower or higher access to food, especially in the stage of the COVID-19 pandemic.

INTERVENTION COMPONENTS

-Apparently the meetings with parents were only informative, since the educational strategy used, the duration of the sessions, the materials used and the evaluation or follow-up carried out with the parents are not detailed.

-It is necessary to describe the pedagogical or psychoeducational strategy used in the nutritional education sessions, how the contents of the 8 sessions were defined? and what was expected to be achieved with each one of them? As well as the evidence of the learning achieved.

- It is necessary to know the messages and materials used to promote dietary diversity, describe how the messages were defined, and how the exposure to information in parents and adolescents was controlled.

- It is necessary to clearly define the variable "dietary diversity".

- The 24-hour recall questionnaire only reports information on the diet of the previous day, so the diversity of the diet cannot be evaluated, or it can be used if it is applied repeatedly during 3 days of the week.

- Considering that there is no cut-off point to indicate adequate or inadequate dietary diversity, it is arbitrary to use the mean score without considering the nutritional quality of the food, the cultural relevance and the accessibility of the food.

Results

- At the end of the intervention, results are presented that indicate a higher mean dietary diversity score in the intervened group compared to the control group (Fig 3), but it is not possible to interpret what these differences would constitute in terms of dietary diversity.

- In the difference in difference (DID) analysis (Table 3), there is speculation of an increase in one unit of dietary diversity in the intervened group, but controlled models are required to verify if these differences persist when correcting for income, the educational level of the caregiver, and assess the effect of the COVID-19 pandemic on food availability and access.

Discussion

The following statement "Our study demonstrated that school-based nutrition education has potential to increase dietary diversity among the adolescent girls", requires that what is "information on nutrition" and "nutritional education" be clearly differentiated; since in this research no an educational intervention was justified.

Conclusion

It is not desirable that the fact of reviewing some nutrition content in school lasting one hour for 8 weeks can be considered a nutritional education strategy, and a public policy be suggested based on it.

6. PLOS authors have the option to publish the peer review history of their article (what does this mean?). If published, this will include your full peer review and any attached files.

Reviewer #1: No

Reviewer #2: No

---

## [Author Response · Author response to Decision Letter 0]

21 Sep 2022

Responses to reviewers’ comments

We appreciate the very thoughtful reviews of the previous version of the manuscript. We have updated the text in response to the reviewers’ queries and feedback. A point-by-point response to each of the reviewers’ comments is included below. We believe these changes have substantially improved the manuscript. We hope you will find this revised manuscript appropriate for publication in PLOS ONE. Many thanks for your consideration.

Reviewer #1: The sample size needs rationale based on intra class correlation in the cluster environment. They state that they calculated sample size based on the effect size of 20 percentage point using the formula of sample size estimation for conventional cluster randomized control trial. They do not state the anticipated power based on this approach.

The main result is seen in Table 3 using the DID which revealed that the mean dietary diversity score among the adolescent girls was likely to be increased by 1 unit due to intervention. The authors give no explanation as to the clinical significance of this difference of one unit. They do, in fact, discuss some limitations of the study seen in the discussion section.

They did state that during performing the DID they adjusted for adolescent’s age, adolescent girls’ father’s age, years of schooling of caregiver, adolescents’ father’s years of schooling, household’s monthly income, household head’s occupation and asset index. This data should be presented as to the significance of these factors, if any.

Response: Thank you for your valuable suggestion. We have stated anticipated power this time in the updated manuscript (mentioned in the “Sample size” part under the “Methods and Materials” section). 

Thank you once again for noting the issue of explanation as to the clinical significance of explanation as to the difference of one unit change in mean dietary diversity score. In this revised manuscript we have explained it under the result section. During performing DID analysis, we adjusted the following variables: adolescent’s age, adolescent girls’ father’s age, years of schooling of caregiver, adolescents’ father’s years of schooling, household’s monthly income, household head’s occupation and asset index in the DID model. As you suggested, we have presented these as supporting information (S2 Appendix) in this revised manuscript. 

Reviewer #2: General comments:

The aim of this study was to measure the efficacy of school-based nutrition education on dietary diversity of the adolescent girls in Bangladesh. The paper is interesting, but the measuring or evaluating the effectiveness of nutrition education is something very compromising, especially because the type of nutrition education or the pedagogical basis are not specified. The execution period of the community trial is striking, which was shortly before the COVID-19 pandemic began and the study ended in the total closure of schools (September 2020), which had an effect on the results, and therefore affects the conclusion reached with this research.

Specific comments:

INTRODUCTION

- In the introduction, "nutritional education" is not defined and the theoretical references of the educational intervention are not established.

- The hypothesis of the investigation is not justified with valid antecedents.

Response: Thank you for your valuable suggestions. We have provided definition of nutrition education with theoretical references and hypothesis of the investigation has been justified in “Introduction” section according to your suggestion, in the revised manuscript.

MATERIALS AND METHODS:

Study design and study population

Within its inclusion criteria, indicators of eating behavior disorders are not mentioned, which is more frequent in adolescent women.

Specify the characteristics of urban and rural schools, and the contexts that could have defined a lower or higher access to food, especially in the stage of the COVID-19 pandemic.

Response: Thank you for your comment. We totally agree with you that eating behavior disorders are more frequent in adolescent women. However, this is one of the limitations of our study that we could not collect data regarding this because we had to make our questionnaire brief and short to finish the interview of the participants (adolescent girls) within a limited time (some particular times from school hours). We had to request teachers to give us some separate times when the interview would be conducted (we mentioned this limitation under the “Discussion” section in the updated manuscript). Our questionnaire was divided into 2 parts: questionnaire for adolescent girls at schools and questionnaire for the mother or caregiver of the index adolescent girl at household level. The questionnaire for adolescent girls consisted of demographic information of the index adolescent girls, dietary diversity, menstrual history and health and nutritional status. 

As you suggested, we have specified the characteristics of urban and rural schools in the revised manuscript. However, our baseline survey was conducted in January 2020 when there was no COVID-19 pandemic situation. As we used the same questionnaire in baseline and endline, so we did not take any data representing the contexts that could have defined a lower or higher access to food, especially in the stage of the COVID-19 pandemic. It is one of the limitations (mentioned in the limitation part of under “Discussion” section).

INTERVENTION COMPONENTS

- Apparently the meetings with parents were only informative, since the educational strategy used, the duration of the sessions, the materials used and the evaluation or follow-up carried out with the parents are not detailed.

Response: Thank you for your comment. An hour-long nutrition education session (60 minutes to 80 minutes long) was provided among the girls of each grade once in a week by the trained female staff who received training on how to impart nutrition education among adolescent girls, from the investigators of the study (it has been mentioned in the “Nutrition education session” part in the updated manuscript). 

As per your suggestions, we have described the materials used and the evaluation carried out in the revised manuscript (it has been mentioned in ‘Nutrition education session and methods and materials section). We have also provided the material- a picture of our poster containing FAO recommended 16 food groups representing dietary diversity as supporting information – S1 Appendix.

- It is necessary to describe the pedagogical or psychoeducational strategy used in the nutritional education sessions, how the contents of the 8 sessions were defined? and what was expected to be achieved with each one of them? As well as the evidence of the learning achieved.

Response: Thank you for your valuable suggestions. We have described how the contents of the 8 sessions were defined in the “Nutrition education session” section under “Materials and methods” of the revised manuscript. We also described pedagogical strategy and the evidence of learning achieved in the revised manuscript.

We conducted quiz session in every week before starting the next session to speculate how much they could understand about the previous session (mentioned in the “Nutrition education session” part in the updated manuscript). Except for the quiz, we did not incorporate any other evidence-generating tool by which we could measure their knowledge about the given intervention, and it is one of our limitations worth to be mentioned (mentioned in the limitation part under the “Discussion” section in the updated manuscript). However, to enhance compliance with the intervention, monitoring of intervention implementation was conducted by supervisory visits to schools and periodic meetings with the school principal, and teachers and by facilitating the peer-education.

- It is necessary to know the messages and materials used to promote dietary diversity, describe how the messages were defined, and how the exposure to information in parents and adolescents was controlled.

Response: As you suggested we have described the messages and materials used to promote dietary diversity in the ‘Nutrition education session’ section and ‘Information, education and communication materials’ section under the heading of ‘Methods and materials. As per your suggestions, we also have described how the exposure to information in parents and adolescents was controlled in the section of study design and study population under the heading of ‘Methods and martials’.

- It is necessary to clearly define the variable "dietary diversity".

Response: Thank you for your valuable suggestion. We have provided definition of dietary diversity in the updated manuscript (mentioned in the “Trial outcome” section of the updated manuscript).

- The 24-hour recall questionnaire only reports information on the diet of the previous day, so the diversity of the diet cannot be evaluated, or it can be used if it is applied repeatedly during 3 days of the week.

Response: Thank you for your comment. We used FAO recommended dietary diversity questionnaire. The recall period of 24 hours was chosen by FAO as it is less subject to recall error, less cumbersome for the respondent, and also conforms to the recall time period used in many dietary diversity studies. Moreover, analysis of dietary diversity data based on a 24-hour recall period is easier than with longer recall period (Reference: Kennedy G, Ballard T, Dop M. Guidelines for Measuring Household and Individual Dietary Diversity. Rome, Italy: Division NaCP; 2010.)

- Considering that there is no cut-off point to indicate adequate or inadequate dietary diversity, it is arbitrary to use the mean score without considering the nutritional quality of the food, the cultural relevance and the accessibility of the food.

Response: We agree with your comment. There is no established cut-off to measure dietary diversity scores for adolescent girls. According to FAO guidelines (Reference: Kennedy G, Ballard T, Dop M. Guidelines for Measuring Household and Individual Dietary Diversity. Rome, Italy: Division NaCP; 2010), some food groups in the dietary diversity questionnaire are combined into a single food group. Following the guidelines, we combined 16 food groups into 9 food groups. Then dietary diversity score was calculated which was normally distributed. For this reason, we calculated the mean score. 

Dietary diversity score (DDS) is defined as a number of individual food groups consumed over a given period of time (FHI F. 360. Minimum dietary diversity for women: a guide for measurement Rome. Italy: FAO; 2016). It reflects the quality of diet at the household or individual level. In addition, DDS is a measure of food security, nutrition information, early warning system and target of intervention at Global or national level [Kennedy G, Ballard T, Dop MC. Guidelines for measuring household and individual dietary diversity: food and agriculture Organization of the United Nations; 2011]. By following FAO’s “Guideline for measuring household and individual dietary diversity, we calculated the dietary diversity score in both the control and intervention arm which reflected the quality of diet at the household or individual level. However, it is one of our limitations that we did not consider the cultural relevance and the accessibility of the food.

Results

- At the end of the intervention, results are presented that indicate a higher mean dietary diversity score in the intervened group compared to the control group (Fig 3), but it is not possible to interpret what these differences would constitute in terms of dietary diversity.

Response: We agree with your comment. We only described our result that mean dietary diversity score was higher among the intervention group than the control group at the endline survey. For presenting the effect of intervention, we performed DID model and in the revised manuscript, we have shown the variables adjusted in the model. 

- In the difference in difference (DID) analysis (Table 3), there is speculation of an increase in one unit of dietary diversity in the intervened group, but controlled models are required to verify if these differences persist when correcting for income, the educational level of the caregiver, and assess the effect of the COVID-19 pandemic on food availability and access. 

Response: Thank you for your valuable comment about the DID analysis. During performing DID analysis, we adjusted the following variables - adolescent’s age, adolescent girls’ father’s age, years of schooling of caregiver, adolescents’ father’s years of schooling, household’s monthly income, household head’s occupation and asset index in the DID model and this has been given as supporting information (S2 Appendix) in the updated manuscript.

Thank you for noting the issue on the effect of the COVID-19 pandemic on food availability and access. Actually, our baseline survey was conducted in January 2020 when there was no COVID-19 pandemic situation. As we used the same questionnaire in baseline and endline, so we did not collect data representing the effect of the COVID-19 pandemic on food availability and access. It is one of our limitations (mentioned in the limitation part of the “Discussion” section).

Discussion

The following statement "Our study demonstrated that school-based nutrition education has potential to increase dietary diversity among the adolescent girls", requires that what is "information on nutrition" and "nutritional education" be clearly differentiated; since in this research no an educational intervention was justified.

Response: Thank you for your valuable suggestions. We have corrected the above-mentioned issues in the second paragraph under “Discussion” section of revised manuscript.

Conclusion

It is not desirable that the fact of reviewing some nutrition content in school lasting one hour for 8 weeks can be considered a nutritional education strategy, and a public policy be suggested based on it.

Response: Thank you for your comment. We wanted to establish a sustainable model of school-based nutrition education for improving dietary diversity among adolescent girls. Scaling up of an innovative intervention and its sustainability is essential for an effective and longer-term impact of the intervention. We found that some previous studies provided food as an intervention among school-going adolescent girls but the withdrawal of direct food under the school feeding program (SFP) is very likely to have negative impact on the sustainability of the program [Moucheraud C, Sarma H, Ha TTT, Ahmed T, Epstein A, Glenn J, et al. Can complex programs be sustained? A mixed methods sustainability evaluation of a national infant and young child feeding program in Bangladesh and Vietnam. BMC Public Health. 2020;20(1):1361. Epub 2020/09/06. doi: 10.1186/s12889-020-09438-2. PubMed PMID: 32887601; PubMed Central PMCID: PMCPMC7487916.]. Our study has focused on behavior change of the adolescent girls through nutrition education at school setting. However, the shorter duration of the intervention in our study could not demonstrate whether it could change behavior of the adolescent girls in increasing dietary diversity through school-based nutrition education in a sustainable manner, rather it showed a pathway on how to increase dietary diversity at school setting.

---

## [Editor Report · Decision Letter 1]

24 Oct 2022

PONE-D-21-39192R1Dietary Diversity Modification through School-Based Nutrition Education among Bangladeshi Adolescent Girls: A Cluster Randomized Controlled TrialPLOS ONE

Dear Dr. Rahman

Thank you for submitting your manuscript to PLOS ONE. After careful consideration, we feel that it has merit but does not fully meet PLOS ONE’s publication criteria as it currently stands. Therefore, we invite you to submit a revised version of the manuscript that addresses the points raised during the review process.

Please submit your revised manuscript by Dec 08 2022 11:59PM If you will need more time than this to complete your revisions, please reply to this message or contact the journal office at plosone@plos.org. Please include the following items when submitting your revised manuscript:A rebuttal letter that responds to each point raised by the academic editor and reviewer(s). You should upload this letter as a separate file labeled 'Response to Reviewers'.A marked-up copy of your manuscript that highlights changes made to the original version. You should upload this as a separate file labeled 'Revised Manuscript with Track Changes'.An unmarked version of your revised paper without tracked changes. You should upload this as a separate file labeled 'Manuscript'.If applicable, we recommend that you deposit your laboratory protocols in protocols.io to enhance the reproducibility of your results. Protocols.io assigns your protocol its own identifier (DOI) so that it can be cited independently in the future. For instructions see: https://journals.plos.org/plosone/s/submission-guidelines#loc-laboratory-protocols. Additionally, PLOS ONE offers an option for publishing peer-reviewed Lab Protocol articles, which describe protocols hosted on protocols.io. Read more information on sharing protocols at https://plos.org/protocols?utm_medium=editorial-email&utm_source=authorletters&utm_campaign=protocols.

We look forward to receiving your revised manuscript.

Kind regards,

Marcos Galván, Ph.D

Guest Editor

PLOS ONE

Journal Requirements:

Additional Editor Comments:

In general, the authors have given a satisfactory response to all the comments made by the reviewers; however, there are some texts that must be clarified or eliminated according to the methodology and scope of the research results.

INTRODUCTION

The definition of nutrition education provided in the introduction is limited but fits the model developed in the intervention. It is suggested on future occasions to include an expert in health education.

TITLE: Dietary Diversity Modification through School-Based Nutrition Education among Bangladeshi Adolescent Girls: A Cluster Randomized Controlled Trial

It does not meet the methodological criteria to be a cluster controlled clinical trial, since there are serious limitations in the research, which have been detailed in the comments made by the reviewers, so it is suggested to modify by: Dietary Diversity Modification through School-Based on Nutrition Knowledge among Bangladeshi Adolescent Girls: A Cluster Randomized Non-Controlled Trial

MATERIALS AND METHODS:

Sample Size: A satisfactory answer has been given; therefore, 150 girls are required for rural schools, and 150 for urban schools, or otherwise control by type of locality. Place this section immediately after randomization.

Apparently there is no cut-off point to establish the diversity of the diet, so the greatest limitation of the study is that the increase of 1 point in the diversity of the diet in the intervened group does not have an interpretation or clinical significance.

DISCUSSION:

The authors must assume a clear position based on the scientific literature on what a point of difference in the diversity of the diet between the intervened and control group means, you must cite other studies that have applied the same scale in other populations.

It remains to argue the role that nutrition education plays within the school curriculum and how it could be used to increase dietary diversity in the long term.

An explanation should be included as to why the revised knowledge in nutrition education had the effect of increasing the consumption of this type of food, such as visor meat and the consumption of legumes, seeds and nuts.

In addition, it must incorporate the current vision of the planetary diet of eat-lancet to reduce the consumption of proteins of animal origin, and what implications it could have on the growth and nutrition of children and adolescents in Bangladesh.

CONCLUSION:

The scope of the nutrition education intervention is limited, since it lacks pedagogical theoretical support and does not have adequate didactics, so it is suggested to eliminate the recommendation to expand the intervention on a national scale.

It should be noted that other elements of the food environment should be included in future studies to improve dietary diversity in adolescent school-age women.

APPENDIX:

Include appendix 1 in English and the original language.

---

## [Author Response · Author response to Decision Letter 1]

22 Dec 2022

We appreciate the very thoughtful reviews of the previous version of the manuscript. We have updated the text in response to the reviewers’ queries and feedback. A point-by-point response to each of the reviewers’ comments is included below. We believe these changes have substantially improved the manuscript. We hope you will find this revised manuscript appropriate for publication in PLOS ONE. Many thanks for your consideration.

Journal Requirements:…………………………………………………………………

Please review your reference list to ensure that it is complete and correct 

If you have cited papers that have been retracted, please include the rationale for doing so in the manuscript text, or remove these references and replace them with relevant current references. Any changes to the reference list should be mentioned in the rebuttal letter that accompanies your revised manuscript. If you need to cite a retracted article, indicate the article’s retracted status in the References list and also include a citation and full reference for the retraction notice.

Response: Thank you for your comment. We have updated the reference list accordingly. 

Additional Editor Comments:………………………………………………………. 

In general, the authors have given a satisfactory response to all the comments made by the reviewers; 

however, there are some texts that must be clarified or eliminated according to the methodology and scope of the research results.

INTRODUCTION

The definition of nutrition education provided in the introduction is limited but fits the model developed in the intervention. It is suggested on future occasions to include an expert in health education.

Response: Thank you for your valuable suggestion. If the intervention is retested, we will recommend including an expert in health education.

Comment on TITLE:

Dietary Diversity Modification through School-Based Nutrition Education among Bangladeshi Adolescent Girls: A Cluster Randomized Controlled Trial

It does not meet the methodological criteria to be a cluster controlled clinical trial, since there are serious limitations in the research, which have been detailed in the comments made by the reviewers, so it is suggested to modify by: Dietary Diversity Modification through School-Based on Nutrition Knowledge among Bangladeshi Adolescent Girls: A Cluster Randomized Non-Controlled Trial.

Response: We agree with you that it doesn’t fully meet the criteria of a cluster-controlled clinical trial. It was a school-based trial, so there were some limitations. 

In our revised manuscript, we described in the ‘Study Design and Study Population” part under the “Methods and Material” section that we tried our best to maintain sufficient buffer zone between control and intervention schools. In the rural setting, the intervention and control schools were located in two completely different sub-districts. In the case of urban schools, even though intervention and control schools were located in Rangpur Sadar (same sub-district), they were located far from one another – the intervention school was located 4.9 km away from the control school.

We also agree that the number of clusters were also limited. However, in support of our research, we would like to mention ha there was another two‑armed cluster randomized controlled trial (RCT) conducted in 10 control and 10 intervention villages in Mangochi, Southern Malawi. In this study, they used intervention similar to us. They provided supplementary nutrition education, dietary counselling and routine ANC services in the intervention group whereas the controls received only routine ANC services (Reference- Katenga-Kaunda, L.Z., Kamudoni, P.R., Holmboe-Ottesen, G. et al. Enhancing nutrition knowledge and dietary diversity among rural pregnant women in Malawi: a randomized controlled trial. BMC Pregnancy Childbirth 21, 644 (2021). https://doi.org/10.1186/s12884-021-04117-5) 

Based upon all these contexts, we request you to consider our title as it is. 

Comment on MATERIALS AND METHODS:

Sample Size: A satisfactory answer has been given; therefore, 150 girls are required for rural schools, and 150 for urban schools, or otherwise control by type of locality. Place this section immediately after randomization.

Response: Thank you for your suggestion. We placed this section after randomization in the updated manuscript. We placed the “Figure 1. Trial flow diagram” (describing clear view of randomization and sample) immediately after “Randomization section” as you suggested.

Comment on MATERIALS AND METHODS

Apparently there is no cut-off point to establish the diversity of the diet, so the greatest limitation of the study is that the increase of 1 point in the diversity of the diet in the intervened group does not have an interpretation or clinical significance.

Response: Thank you for your valuable comment. We agree with you that the increase of 1 point in the diversity of the diet in the intervened group does not have clinical significance. We have included this as one of the limitations under the limitation part of discussion section in the revised manuscript. However, although mean dietary diversity score was increased only 1 point in intervention arm compared to control arm, this increase was statistically significant after all other variables were adjusted. Moreover, our intervention on school-based nutrition education increased the mean score of dietary diversity that could be minimal, but it was significant within a shorter period and with many challenges, and the results indicated increase of some food groups in the intervention group which are – “other vitamin A rich fruits and vegetables”, “organ meat”, “legumes, nuts and seeds” and “milk and milk products”.

Comment on DISCUSSION: The authors must assume a clear position based on the scientific literature on what a point of difference in the diversity of the diet between the intervened and control group means, you must cite other studies that have applied the same scale in other populations.

Response: Thank you for your comment. We would like to mention that dietary diversity score was determined by adding the number of food categories consumed by each participant over the previous 24 hours. The questionnaire contained 16 food groups and we analyzed 9 food groups because we combined certain food groups into a single food group for analytical purpose (according to FAO guideline). 

Our study population was 10-14 years of adolescent girls. According to FAO’s “Guidelines for measuring household and individual dietary diversity”, there is no established cut-off point to measure dietary diversity for adolescent girls of 10-14 years of age (Reference number 27 in the revised). The guideline set cut-off of minimum dietary diversity for 15-49 years of women and some other studies worked on “improving dietary diversity” followed this cut-off as their study participants age followed 15-49 years of age ((Reference number 26, 30 in the revised manuscript) and also another study was done in Malawi - (Reference- Katenga-Kaunda, L.Z., Kamudoni, P.R., Holmboe-Ottesen, G. et al. Enhancing nutrition knowledge and dietary diversity among rural pregnant women in Malawi: a randomized controlled trial. BMC Pregnancy Childbirth 21, 644 (2021). https://doi.org/10.1186/s12884-021-04117-5)). 

Comment on DISCUSSION: It remains to argue the role that nutrition education plays within the school curriculum and how it could be used to increase dietary diversity in the long term.

Response: Thank you for your concern. Alike our intervention study, some studies conducted in similar settings indicated that along with the regular curriculum in secondary schools, nutrition education can be added on to improve the dietary diversity of adolescent girls [Reference number -26,29,30 in manuscript]. They suggested that school-based nutrition and health education should be a part of comprehensive school health programs to reach both the students and their families [Reference number - 26 in the revised manuscript].

In low- and middle-income countries, the school feeding program (SFP) has been found instrumental to increasing dietary diversity of adolescent girls but such a program has some limitations including irregular supply and storage of materials (i.e., food items) and loss of educational time at school [Reference number - 23]. Withdrawn of direct food under the school feeding program (SFP) is very likely to have a negative impact on the sustainability of the program [Reference number - 33]. In our study, unlikely to school feeding program, we used existing materials of schools like – laptops, projectors. So, it can be said that, this kind of intervention has high chance to be sustainable in long run. Our trained staffs provided nutrition education sessions in our study. To maintain sustainability, short-term teacher’s training of existing school is enough to deliver the nutrition education session. (We have updated these parts in our “Discussion” section in the updated manuscript). 

Comment on DISCUSSION: An explanation should be included as to why the revised knowledge in nutrition education had the effect of increasing the consumption of this type of food, such as visor meat and the consumption of legumes, seeds and nuts.

Response: Thanks. We have included the explanation as follows in the revised manuscript:

Due to our intervention, consumption of some food groups was increased among intervention arm than in control arm in endline – “other vitamin A rich fruits and vegetables”, “organ meat”, “legumes, nuts and seeds” and “milk and milk products”. Parents’ meetings and nutrition education sessions may be the main reasons of improving the consumption of some food groups. 

During parent’ meeting, we discussed about the necessity of dietary diversity for young adolescent girls and we described parent’s role in improving dietary diversity by buying and cooking diversified foods for their family members. We provided posters containing 16 food groups representing dietary diversity among adolescent girls and suggested them to hang the poster in the kitchen where their caregivers cook or in front of dining table where all family members gather to eat to practice dietary diversity in everyday life (We highlighted these parts in our “Intervention components” part for your kind reading). 

During our nutrition education sessions, we provided detailed description of 16 food groups by showing locally available Bangladeshi food items pictures. We also informed them how to choose diversified diets in low cost (We highlighted these parts in our “Intervention components” part for your kind reading). Students shared with us that they did not know about some food groups which should be eaten for the betterment of health like – organ meat. Lack of knowledge was the main problem to not practicing the diversified food among adolescent girls. That’s why EAT – Lancet Commission also supported that individuals should be educated on healthy diet (Reference no 41 in our updated manuscript) 

Comment on DISCUSSION: In addition, it must incorporate the current vision of the planetary diet of eat-lancet to reduce the consumption of proteins of animal origin, and what implications it could have on the growth and nutrition of children and adolescents in Bangladesh.

Response: Thank you for your valuable suggestion. We have incorporated a discussion part under discussion section in the updated manuscript according to your suggestion. 

Comment on CONCLUSION: The scope of the nutrition education intervention is limited, since it lacks pedagogical theoretical support and does not have adequate didactics, so it is suggested to eliminate the recommendation to expand the intervention on a national scale. It should be noted that other elements of the food environment should be included in future studies to improve dietary diversity in adolescent school-age women.

Response: Thank you for your valuable comments. As you suggested we have revised conclusions (page 23) section of the manuscript as follows:

‘The shorter duration of the intervention in our study could not demonstrate whether it could change the behavior of adolescent girls in increasing dietary diversity through school-based nutrition education, instead it showed a pathway on how to increase dietary diversity at school setting. With a limited resource, our school-based nutrition education intervention in real settings of the schools has demonstrated its feasibility in improving dietary diversity of the adolescent girls. Since this study was conducted in the COVID-19 situation with many other challenges and limitations it recommends to retest this intervention before scaling up in the larger scale. In retesting, we recommend including more clusters and other elements of the food environment to increase its precision and acceptability.’

Comment on APPENDIX: Include appendix 1 in English and the original language.

Response: Thank you for your comment. We followed the FAO recommended 16 food group chart which is available in the “Guidelines for Measuring Household and Individual Dietary Diversity” (it is our reference number 27) and considered locally available Bangladeshi foods for developing the poster. In the revised manuscript, we have provided the English one under Appendix-1, indicating with the numbers that present different food groups.

---

## [Editor Report · Decision Letter 2]

5 Jan 2023

PONE-D-21-39192R2Dietary Diversity Modification through School-Based Nutrition Education among Bangladeshi Adolescent Girls: A Cluster Randomized Controlled TrialPLOS ONE

Dear Dr. Rahman,

Thank you for submitting your manuscript to PLOS ONE. After careful consideration, we feel that it has merit but does not fully meet PLOS ONE’s publication criteria as it currently stands. Therefore, we invite you to submit a revised version of the manuscript that addresses the points raised during the review process. Please submit your revised manuscript by Feb 19 2023 11:59PM. If you will need more time than this to complete your revisions, please reply to this message or contact the journal office at plosone@plos.org. Please include the following items when submitting your revised manuscript:A rebuttal letter that responds to each point raised by the academic editor and reviewer(s). You should upload this letter as a separate file labeled 'Response to Reviewers'.A marked-up copy of your manuscript that highlights changes made to the original version. You should upload this as a separate file labeled 'Revised Manuscript with Track Changes'.An unmarked version of your revised paper without tracked changes. You should upload this as a separate file labeled 'Manuscript'.If applicable, we recommend that you deposit your laboratory protocols in protocols.io to enhance the reproducibility of your results. Protocols.io assigns your protocol its own identifier (DOI) so that it can be cited independently in the future. For instructions see: https://journals.plos.org/plosone/s/submission-guidelines#loc-laboratory-protocols. Additionally, PLOS ONE offers an option for publishing peer-reviewed Lab Protocol articles, which describe protocols hosted on protocols.io. Read more information on sharing protocols at https://plos.org/protocols?utm_medium=editorial-email&utm_source=authorletters&utm_campaign=protocols.

We look forward to receiving your revised manuscript.

Kind regards,

Marcos Galván, Ph.D

Guest Editor

PLOS ONE

Journal Requirements:

Additional Editor Comments (if provided):

The authors have made the requested corrections. The main document is attached with some very specific comments, and it is suggested to update the conclusion in the abstract.
---

## [Author Response · Author response to Decision Letter 2]

2 Feb 2023

Response to reviewers’ comment

Thank you for your valuable comment. Response to the comments is included below: 

Reviewer comment: The authors have made the requested corrections. The main document is attached with some very specific comments, and it is suggested to update the conclusion in the abstract.

Response: Thank you for your valuable comments. As you suggested we have updated the inclusion in the abstract in the revised version of the manuscript.

---

## [Editor Report · Decision Letter 3]

15 Feb 2023

Dietary Diversity Modification through School-Based Nutrition Education among Bangladeshi Adolescent Girls: A Cluster Randomized Controlled Trial

PONE-D-21-39192R3

Dear Dr. Rahman,

We’re pleased to inform you that your manuscript has been judged scientifically suitable for publication and will be formally accepted for publication once it meets all outstanding technical requirements.

Kind regards,

Marcos Galván, Ph.D

Guest Editor

PLOS ONE

Additional Editor Comments (optional):

The authors have made the requested corrections, so it is accepted.
---

## [Editor Report · Acceptance letter]

28 Feb 2023

PONE-D-21-39192R3 

Dietary Diversity Modification through School-Based Nutrition Education among Bangladeshi Adolescent Girls: A Cluster Randomized Controlled Trial 

Dear Dr. Rahman:

I'm pleased to inform you that your manuscript has been deemed suitable for publication in PLOS ONE. Congratulations! Your manuscript is now with our production department. 

Kind regards, 

on behalf of

Dr. Marcos Galván 

Guest Editor

PLOS ONE